# Effects of speed, agility and quickness training programme on cognitive and physical performance in preadolescent soccer players

**Athos Trecroci**[1], **Luca Cavaggioni**[1,2], **Alessio Rossi**[3], **Andrea Moriondo**[4], **Giampiero Merati**[5,6], **Hadi Nobari**[7,8,9], **Luca Paolo Ardigò**[10,11]✉*, **Damiano Formenti**[5]✉

**1** Department of Biomedical Sciences for Health, Università degli Studi di Milano, Milano, Italy, **2** Department of Endocrine and Metabolic Diseases, Obesity Unit and Laboratory of Nutrition and Obesity Research, IRCCS Istituto Auxologico Italiano, Milan, Italy, **3** Department of Computer Science, University of Pisa, Pisa, Italy, **4** Department of Medicine and Surgery, School of Medicine, University of Insubria, Varese, Italy, **5** Department of Biotechnology and Life Sciences, University of Insubria, Varese, Italy, **6** IRCCS Fondazione don Carlo Gnocchi, Milano, Italy, **7** Department of Exercise Physiology, Faculty of Educational Sciences and Psychology, University of Mohaghegh Ardabili, Ardabil, Iran, **8** Department of Motor Performance, Faculty of Physical Education and Mountain Sports, Transilvania University of Braşov, Braşov, Romania, **9** Faculty of Sport Sciences, University of Extremadura, Cáceres, Spain, **10** Department of Teacher Education, NLA University College, Oslo, Norway, **11** Department of Neurosciences, Biomedicine and Movement Sciences, School of Exercise and Sport Science, University of Verona, Verona, Italy

☯ These authors contributed equally to this work.
* luca.ardigo@univr.it

**Data Availability Statement:** All relevant data are within the manuscript and its Supporting Information files.

## Abstract

The aim of the present study was to investigate the effect of a short-term (4 weeks) non-soccer-specific training programme based on speed, agility and quickness (SAQ) and a soccer-specific training programme based on small-sided games (SSG) on cognitive and physical performance in preadolescent soccer players. Twenty-one participants were randomly assigned to SAQ group (n = 11) or SSG group (n = 10). They were tested pre and post interventions on physical (5 m sprint, 20 m sprint and sprint with turns of 90˚) and cognitive (inhibitory control by means of the Flanker task and perceptual speed by means of the visual search task) performances. Although no significant time x group interactions were observed, the main effect of time was significant for cognitive performance and 5 m and 20 m sprint, showing improvements after both SAQ and SSG. These findings highlight that 4 weeks of SAQ training programme induced comparable improvements in cognitive and physical performance with respect to a soccer-specific training programme based on SSG in preadolescent soccer players. Non-sport-specific activities targeting speed, agility and quickness combined with cognitive engagement (i.e., SAQ) should be useful strategies as soccer-specific activities to be included within a soccer training programme for promoting both physical and cognitive domain in preadolescent individuals.

## Introduction

Soccer requires an extraordinary balance between physiological performance, motor control and mental and cognitive abilities [1]. While the game is developed in an unpredictable and

**Funding:** This work is supported by the European Community's H2020 Program under the Funding Scheme H2020-INFRAIA-2019-1 Research Infrastructures Grant Agreement 871042, www.sobigdata.eu, accessed on 2 November 2021, SoBigData++: European Integrated Infrastructure for Social Mining and Big Data Analytics. The funders had no role in study design, data collection and analysis, decision to publish, or preparation of the manuscript.

**Competing interests:** The authors have declared that no competing interests exist.

changing context, players must process several information in a short time under a stressful condition with the aim of anticipating, planning and executing appropriate motor actions [1]. The related underlying mental construct of this aspect is defined as perceptual-cognitive skills, which refer to the players' abilities of selecting and perceiving information from the environment [2]. In this context, a match requires players able to execute soccer-specific motor actions (such as passing, dribbling and shooting) under high perceptual-cognitive demands [3]. Accordingly, a recent but remarkable body of literature on the cognitive-component skill approach provided evidence that high-performance level athletes perform better in general cognitive tasks (assessing cognitive functions) compared with low-performance level athletes [4–7]. These findings suggest that non-sport-specific cognitive skills such as cognitive control may be related to performance, especially in team sports.

The well-recognized importance of cognition for soccer performance can be also found in the concept of agility, defined as "skills and abilities needed to change direction, velocity or mode in response to a *stimulus*" [8, 9]. As a fundamental determinant of performance in team sports, improving agility through suitable training strategies is mandatory for coaches and practitioners [10–12]. This is particularly important for young players that would benefit from a balanced development of both general and sport-specific *stimuli* especially during the sampling years (6–12 years). In this sense, activities based on non-sport-specific *stimuli* (for example without the ball in soccer) may help to improve physical fitness in individuals regularly practicing a specific sport [13, 14]. For example, non-sport-specific drills can stimulate important determinants for fitness in soccer such as speed, agility and quickness (SAQ [12, 15]). The SAQ training method refers to a training approach based on movement tasks performed with high rate in short-time (quickness) combined with straight (speed) and multidirectional sprints (change of direction, COD) over a variety of distances with and without cognitive *stimuli* (agility [12]). Numerous studies demonstrated the positive effects of the SAQ training on agility and soccer-related performance in young adults, adolescents and preadolescents soccer players [12, 16–18]. Moreover, besides the soccer-related literature, improvements in agility were also found in response to a 4-week training programme with foot-speed and choice-reaction agility drills in active men and women [19].

Apart from physical benefits of agility-related exercises, there are also evidences of their positive effects on brain structures and cognition [20–22]. According to an exercise-cognition conceptual framework, the cognitive engagement required to manage complex physical tasks may be the responsible of the cognitive improvements following specific exercises [21, 22]. In fact, physical exercises including some form of cognitive engagement such as team sports, neuromotor and agility exercises were found to induce beneficial effects on cognition [21–27]. In this scenario, emphasis has been devoted on the cognitive benefits related to physical activity and physical exercises in the context of developing and maintaining health across the whole lifespan [21, 25, 27]. However, also the sport performance literature has begun to address the exercise-cognition interaction by examining the effects of cognitively challenging exercises on both physical and cognitive performance [20]. Accordingly, Lenneman et al. [20] demonstrated that a 6-week agility training programme improved memory and sustained vigilance by ~ 11% and ~ 2%, respectively, whereas no improvement was found after a 6-week running training programme. Furthermore, Lenneman et al. [20] reported similar benefits on selective attention when compared with running. This evidence suggests that exercises aimed at stimulating agility might be a useful strategy for targeting both physical and cognitive performances. It is worth noting that the study by Lenneman et al. [20] was performed in a military setting on adult individuals, whereas whether a training programme based on speed, quickness and agility drills would induce positive benefits on both physical and cognitive domains in preadolescents remains unclear.

We decided to examine the effects of SAQ compared with sport-specific training (i.e., small-sided games, SSG) on physical and cognitive performances in young soccer players. SSG are typically described as smaller versions of the formal game, mostly used to optimize the time of training stimulating the sport-specific determinants (among which physiological, physical, technical, tactical, and cognitive) [28]. From a practical viewpoint, understanding the effect of specific training methodologies on both physical and cognitive domains might be helpful for coaches and practitioners to stimulate qualities related to performance and for a balanced growth targeting both physical and cognitive domains of preadolescent individuals. Therefore, the aim of the present study was to investigate the effect of a short-term (4 weeks) SAQ training programme as compared to a soccer-specific training programme on cognitive and physical performance in preadolescent soccer players. Based on results detected in different populations regarding the association between physical and cognitive abilities [4–7] and–at least as preliminary findings–the mutual beneficial effects of physical and cognitive training [7, 12, 20, 22, 25], we hypothesized that training based on SAQ would be at least as effective as a mere sport-specific training for improving cognitive and physical performance in preadolescent soccer players.

## Materials and methods

### Participants

To assess the effect of a short-term (i.e., 4 weeks) SAQ training programme on cognitive and physical performances of preadolescent soccer players, a randomized pre-post parallel group design was employed. Sample size was not chosen according to sampling theory due to study's novelty. Twenty-one preadolescent soccer players were recruited from one soccer academy in northern Italy and voluntarily participated in the study. They were randomly assigned to either a SAQ training group (SAQ group; n = 11; age 9.7 ± 0.4 yrs (range 9–11 yrs); height 1.34 ± 0.07 m, body mass 32.3 ± 0.6 kg; mean±standard deviation) or a SSG training group (SSG group; n = 10; age 9.5 ± 0.6 yrs (range 9–11 yrs); height 1.34 ± 0.05 m, body mass 32.4 ± 0.5 kg). All participants were accustomed to regular soccer training as part of the weekly routine with three training sessions (lasting 2 h) and a match per week. Players, their parents or legal guardians were informed about the purpose and of the study before giving written informed consent to participate. The ethic committee of the local university approved the study that was conducted in accordance with the Declaration of Helsinki.

### Design

Two experimental testing sessions were scheduled within one week before the start of the training period (pre) and after 4 weeks of training (post). The first testing session aimed at collecting anthropometric variables and at assessing cognitive performance using two cognitive tests assessing inhibitory control and perceptual speed. The second testing session aimed at assessing physical performance as acceleration, sprint and COD performance. After the four weeks of training (namely during the first week thereafter), participants underwent post-test sessions that were identical to the pre-test. Thus, the whole study lasted 6 weeks. Participants were instructed to abstain from strenuous physical activity in the two days before the testing sessions. Before the experimental testing sessions, participants underwent a familiarization session to get accustomed with the testing procedures. An overview of the experimental protocol is shown in Fig 1.

### Cognitive performance

Cognitive performance assessment included two computer-based tasks reflecting inhibitory control (Flanker task) and perceptual speed (visual search task) that were proposed to the

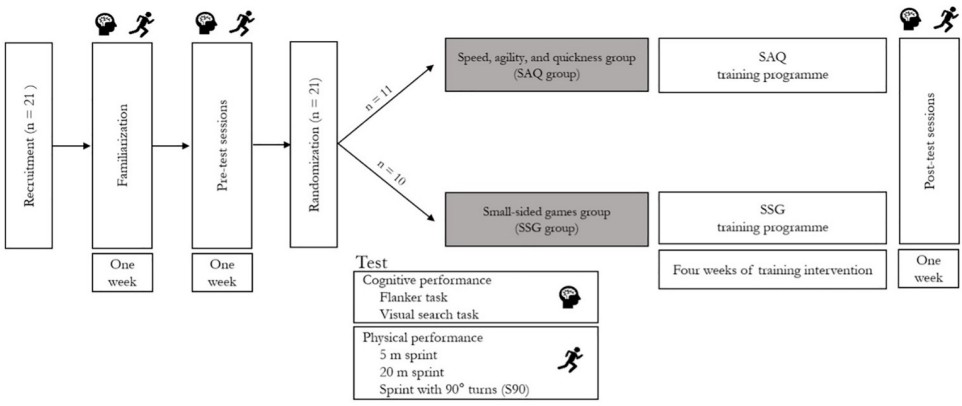

**Fig 1. Overview of the experimental protocol.**

participants in a random order. The two tasks were previously used to assess cognitive performance in young volleyball players [23, 29] and were designed using the software Psytoolkit [30, 31]. Participants were comfortably sat in front of a computer with keyboard in a quiet and isolated room.

## Flanker task

Inhibitory control was assessed using a modified version of the Flanker task with arrows [32]. The subjects were requested to respond as quickly and accurately as possible to the direction of a left or right target arrow while ignoring two flanking arrows on each side pointing in the same or the opposite direction. The task included two different conditions: congruent and incongruent. The congruent condition consisted of trials in which both the target arrow and the four flanking arrows pointed in the same direction (left: $< < < < <$ or right: $> > > > >$). The incongruent condition consisted of trials in which flanking arrows pointed in the opposite direction with respect to the target arrow ($< < > < <$ or $> > < > >$). Participants had to press the button A of the keyboard when the target arrow pointed to the left (i.e., $<$), and the button L when the target arrow pointed to the right (i.e., $>$). For each condition, 100 trials were presented randomly with right and left target arrows as well as congruent and incongruent conditions occurring with the same probability. Participants had 2 s to provide their response to the target arrow. Mean response time of the correct responses was computed for each condition and considered as outcome.

## Visual search task

Perceptual speed was assessed by means of the visual search task [33]. The target *stimulus* was an orange letter T and distractors *stimuli* were blue T and an upside-down orange T. Participants were requested to press the space bar of the keyboard when the target *stimulus* was present among distractor *stimuli* and to avoid a response when the target was absent. For each trial, the numbers of items among which target *stimulus* could be present were 5, 10, 15 and 20, which were randomized across a total of 100 trials. For each item trial, 25 trials were presented randomly, among which trials with the target present or absent were randomized. Participants had to respond as quickly and accurately as possible within 4 s from the trial presentation. Only correct responses were included in the outcome variables. Mean response time of the correct responses was computed for each item trials and considered as outcome.

## Physical performance

Physical performance assessment comprised tests assessing acceleration (5 m sprint), speed (20 m sprint) and COD ability with 90˚ turns (COD90). All tests were randomly performed in a gym with rubber surface at the same time of day (from 5 to 7 p.m.) in both pre and post testing sessions. A 10-min warm-up period consisting of general running exercises (e.g., jogging in different directions) and dynamic stretching preceded the immediate initiation of the testing session. Ten min of rest (followed by a 5-min rewarm-up) were given among tests to ensure a full recovery. Performance time was recorded using a timing gate system (Witty, Microgate, Bolzano, Italy). In line with participants' (limited) ~10-yr age height and allowing a usual a little bit crouched posture during initial acceleration, the timing gates were placed at only 0.60 m above the ground. Participants began the tests from the starting line, that was positioned behind 0.30 m of the pair of photocells. They performed three trials separated by a recovery of 3 min. The best performance time was considered for the analysis.

## Five-m and twenty-m sprint

On command, participants performed three trials of 5-m and 20-m sprint starting from a standing position. They were requested to accelerate from the starting line and to run as fast as possible until the end line. After the end of a trial, participants were asked to return to the starting line by walking slowly.

## Sprint with 90˚ turns

Validity and reliability of COD90 were previously reported [11]. Namely, Sporis et al. found out COD90 reliability to be the highest over six different agility tests [11]. The players began with both feet behind the starting line (point A). After a signal, they were requested to run as fast as possible to point B and make a 90˚ turn to the right. They continued to run to the point C, where they made a 90˚ turn to the left running to the point D. They made another 90˚ turn to the left and ran to the point E, where they made a 90˚ turn to the right reaching the point F. At point F, they made another 90˚ turn to the right running to the point G. At point G, they made a 90˚ turn to the left and ran to the finish line (point H). A schematic representation of the test is shown in Fig 2.

## Training interventions

The experimental protocol was composed by 2 interventions *per* week over 4 weeks (8 sessions, lasting about 25 min each) during the competitive season. The regular week included 3 training sessions (lasting 90 min each) and a match-play *per* week. The training interventions of SAQ group and SSG group were administered at the beginning of the first 2 weekly training sessions (Monday and Wednesday) after a 10-min warm-up with running and dynamic stretching [34]. Then, both groups continued their regular training programme with soccer-related drills (such as dribbling, passing and shooting drills) and game formats. In the third weekly training session (Friday), the two groups performed the same soccer-specific contents [35]. This weekly experimental setting was previously adopted and permitted to maintain a high ecological validity [12, 34, 35]. Each training session occurred at the same time of day (from 5 to 7 p.m.).

The SAQ training programme consisted of a combination of training elements based on brief efforts in the form of SAQ drills. Two phases (each lasting about 10 min with a 5 min of rest between them) of SAQ drills were administered and matched each other for number of drills, work volume/drill and rest between drills. The number of drills, training volume and

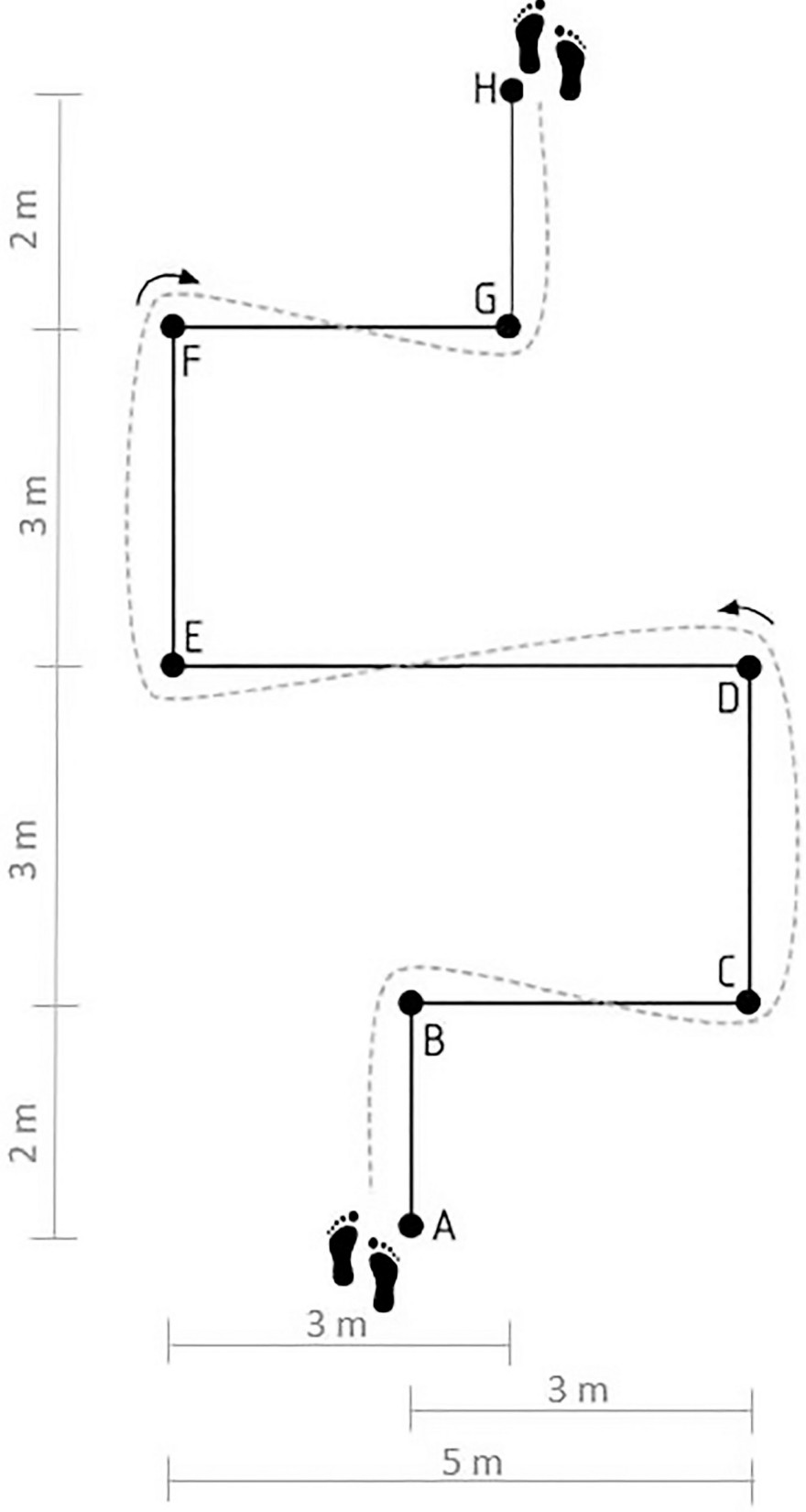

**Fig 2. Scheme of the sprint with 90° turns (COD90).**

rest periods were similar to frameworks previously adopted [12, 14]. Training volume and exercise intensity progressively increased throughout the 4 weeks by manipulating the number of change of directions and sprints. Moreover, the cognitive involvement of each drill was progressively increased from less (few *stimuli*) to more cognitive demanding contexts (i.e., agility drills with multiple *stimuli*).

The SSG training programme was mainly based on technical exercises (passing, shooting and dribbling skills) and on a combination of offensive and defensive game-based drills (i.e., SSG). As for SAQ group, SSG training contents were arranged in two phases lasting 10 min each with 5 min of rest in between. In the first phase, participants underwent the technical exercises, whereas in the second phase they engaged in offensive and defensive game-based drills (including both sprints and changes of direction). The game-based drills were arranged with and without goalkeeper by increasing the number of players throughout the 4-week period.

Borg's rate of perceived exertion scale (running from 0 to 10) was used to ensure both SAQ and SSG training programmes elicited comparable training load session. The training load of each SAQ and SSG session was comparable and ranged from 75 arbitrary units to 88 arbitrary units throughout the experimental periods [36]. Verbal encouragement was provided over both training interventions to encourage the participants throughout the activities.

Table 1 summarizes the training contents of the training interventions of both SAQ and SSG group.

## Statistical analysis

Data are shown as mean±standard deviation. The normality of the distribution of the data was assumed by using the Shapiro-Wilk normality test and visual inspection permitting to perform a parametric statistic. No significant difference between groups–as detected by means of unpaired Student's *t*-test–was found for each variable in pretraining test evaluation confirming

**Table 1. Training content of the 4-week intervention performed by the SAQ group and by the SSG group.**

| Week | SAQ training programme (SAQ group) | SSG training programme (SSG group) |
|---|---|---|
| 1 | Basic footwork exercises (split-steps, line drills, lateral line and multiple hops) with no equipment followed by brief linear sprints over 5 m also combined with cognitive *stimuli* (e.g., auditory signals as for stop and go running drills). | Passing drills |
| | | 1 *versus* 1, 10 x 5 m |
| | | 2 *versus* 1, 12 x 8 m |
| | | Rules: free play without goalkeepers and no coach's encouragement |
| 2 | Basic footwork exercises (skipping, hopscotch and in&out drills) over the speed-ladder followed by brief sprints with 1–3 change of directions at 30˚ and 45˚ over 10 m also combined with cognitive *stimuli* (visual stimuli on fixed targets as for coloured cone running drills). | Shooting drills |
| | | 2 *versus* 1, 12 x 8 m |
| | | 2 *versus* 2, 15 x 10 m |
| | | Rules: free play without goalkeepers and no coach's encouragement |
| 3 | Advanced footwork exercises (foot exchange, icky shuffle and hip twist) over the speed-ladder followed by brief sprints with 3–5 change of directions at 30˚, 45˚ and 90˚ over 10 m also combined with cognitive *stimuli* (combination of auditory signals and visual *stimuli* on fixed targets). | Dribbling drills |
| | | 2 *versus* 2, 15 x 10 m |
| | | 3 *versus* 2, 18 x 12 m |
| | | Rules: few numbers of touches (i.e., 3) with goalkeepers and with coach's encouragement |
| 4 | Combination of basic and advanced footwork exercises with basic and advanced agility drills in response to multiple cognitive *stimuli* (visual *stimuli* on moving targets as for chasing runs and mirror drills) over 15 m. | Combination of technical skills |
| | | 3 *versus* 2, 18 x 12 m |
| | | 3 *versus* 3, 20 x 15 m |
| | | Rules: few numbers of touches (i.e., 2 and 3) with goalkeepers and with coach's encouragement |

the appropriateness of initial random splitting all participants into the two intervention groups. A two-way (time and group) ANOVA with repeated measures on one factor (time) was used to investigate the effect of training intervention on each variable. In case of significant main effects or interactions, the Bonferroni *post-hoc* tests were used as pairwise comparisons. The effect sizes (ES) were calculated to assess the magnitude of the pre-to-post effect using the equation for paired data proposed by Dankel and Loenneke [37]. ES values between 0.20 and 0.49 indicated a *small* ES, values between 0.50 and 0.79 indicated a *medium* ES, and values of 0.80 and above indicated a *large* ES. The statistical significance was set at $P < 0.05$. Statistical analysis was performed using GraphPad Prism version 8.00 for Windows (GraphPad Software, San Diego, CA).

## Results

### Cognitive performance

The effect of training intervention on inhibitory control is shown in Fig 3. Regarding the reaction time in the congruent condition of the Flanker task, no significant interaction (time x group) ($F_{1,19} = 0.22$, $P = 0.64$) or main effect of group ($F_{1,19} = 2.05$, $P = 0.16$) were found, whereas a significant main effect of time was revealed ($F_{1,19} = 10.47$, $P = 0.004$). Specifically, Bonferroni *post-hoc* analysis revealed that the SAQ group significantly improved the reaction time in the congruent condition of the Flanker task from pre to post ($P = 0.029$; ES = 1.10, *large*), whereas the SSG group change was not significant even if *medium* effect size was reported ($P = 0.14$; ES = 0.48, *small*).

Regarding the reaction time in the incongruent condition of the Flanker task, no significant interaction (time x group; $F_{1,19} = 1.53$, $P = 0.23$) or main effect of group ($F_{1,19} = 0.92$, $P = 0.34$) were found, whereas a significant main effect of time was revealed ($F_{1,19} = 10.12$, $P = 0.004$). Specifically, Bonferroni *post-hoc* analysis revealed that the SAQ group significantly improved the reaction time in the congruent condition of the Flanker task from pre to post ($P = 0.009$; ES = 0.96, *large*), whereas the SSG group change was not significant even if *large* effect size was reported ($P = 0.39$; ES = 0.42, *small*).

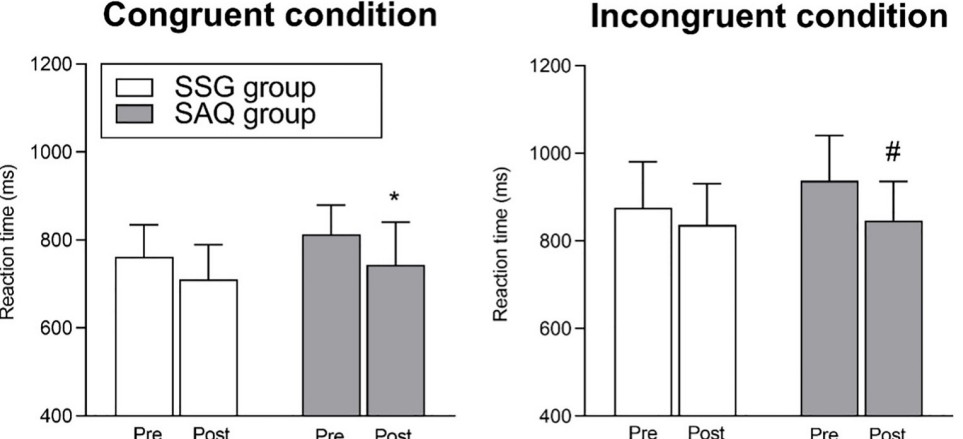

**Fig 3. Effect of training intervention on inhibitory control (reaction time for congruent and incongruent condition in the Flanker task).** Post < pre as from Bonferroni *post-hoc* tests: * $P < 0.05$; # $P < 0.01$.

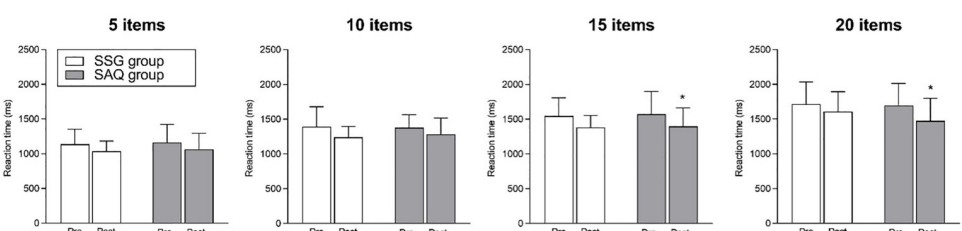

**Fig 4. Effect of training intervention on perceptual speed (reaction time for 5, 10, 15 and 20 items conditions in the visual search task).** Post < pre by Bonferroni post-hoc tests: * P < 0.05.

The effect of training intervention on perceptual speed is shown in Fig 4. Regarding the reaction time of the 5 items of the visual search task, no significant interaction (time x group; $F_{1,19} = 0.005$, $P = 0.44$) or main effect of group ($F_{1,19} = 0.07$, $P = 0.78$) were found, whereas a significant main effect of time was revealed ($F_{1,19} = 9.99$, $P = 0.005$). However, Bonferroni *post-hoc* analysis revealed that neither the SAQ group ($P = 0.07$; ES = 0.63, *medium*) nor the SSG group significantly improved the reaction time of the 5 items of the visual search task from pre to post ($P = 0.07$; ES = 0.76, *medium*).

Regarding the reaction time of the 10 items of the visual search task, no significant interaction (time x group; $F_{1,19} = 0.33$, $P = 0.56$) or main effect of group ($F_{1,19} = 0.02$, $P = 0.86$) were found, whereas a significant main effect of time was revealed ($F_{1,19} = 7.21$, $P = 0.014$). However, Bonferroni *post-hoc* analysis revealed that neither the SAQ group ($P = 0.28$; ES = 0.46, *small*) nor the SSG group significantly improved the reaction time of the 10 items of the visual search task from pre to post ($P = 0.07$; ES = 0.70, *medium*).

Regarding the reaction time of the 15 items of the visual search task, no significant interaction (time x group; $F_{1,19} = 0.02$, $P = 0.86$) or main effect of group ($F_{1,19} = 0.03$, $P = 0.85$) were found, whereas a significant main effect of time was revealed ($F_{1,19} = 12.64$, $P = 0.002$). Specifically, Bonferroni *post-hoc* analysis revealed that the SAQ group significantly improved the reaction time of the 15 items of the visual search task from pre to post ($P = 0.02$; ES = 1.25, *large*), whereas the SSG group change was not significant ($P = 0.06$; ES = 0.57, *medium*).

Regarding the reaction time of the 20 items of the visual search task, no significant interaction (time x group; $F_{1,19} = 0.77$, $P = 0.38$) or main effect of group ($F_{1,19} = 0.41$, $P = 0.52$) were found, whereas a significant main effect of time was revealed ($F_{1,19} = 6.71$, $P = 0.017$). Specifically, Bonferroni *post-hoc* analysis revealed that the SAQ group significantly improved the reaction time of the 20 items of the visual search task from pre to post ($P = 0.04$; ES = 1.47, *large*), whereas the SSG group change was not significant ($P = 0.50$; ES = 0.28, *small*).

## Physical performance

The effect of training intervention on 5 m sprint, 20 m sprint and COD90 is shown in Fig 5.

Regarding the 5 m sprint performance, no significant interaction (time x group; $F_{1,19} = 0.31$, $P = 0.58$) or main effect of group ($F_{1,19} = 0.40$, $P = 0.53$) were found, whereas a significant main effect of time was revealed ($F_{1,19} = 13.18$, $P = 0.001$). Specifically, Bonferroni *post-hoc* analysis revealed that the SAQ group significantly improved the 5 m sprint performance from pre to post ($P = 0.013$; ES = 0.96, *large*), whereas the SSG group change was not significant ($P = 0.09$; ES = 0.63, *medium*).

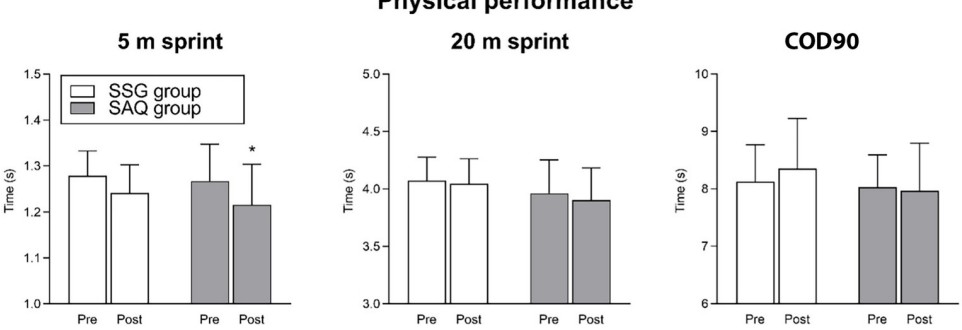

**Physical performance**

**Fig 5. Effect of training intervention on perceptual speed (reaction time for 5, 10, 15 and 20 items conditions in the visual search task).** Post < pre by Bonferroni *post-hoc* tests: * $P < 0.05$.

Regarding the 20 m sprint performance, no significant interaction (time x group; $F_{1,19} = 0.60$, $P = 0.44$) or main effect of group ($F_{1,19} = 1.34$, $P = 0.26$) were found, whereas a significant main effect of time was revealed ($F_{1,19} = 5.21$, $P = 0.034$). However, Bonferroni *post-hoc* analysis revealed that neither the SAQ group ($P = 0.07$; ES = 0.85, *large*) nor the SSG group ($P = 0.62$; ES = 0.27, *small*) significantly improved the 20 m sprint performance from pre to post.

Regarding COD90, no significant interaction (time x group) was revealed ($F_{1,19} = 0.36$, $P = 0.87$). Similarly, the main effect of group ($F_{1,19} = 0.72$, $P = 0.40$) and the main effect of time ($F_{1,19} = 0.30$, $P = 0.59$) were not significant.

## Discussion

The aim of this study was to investigate the effect of a short-term (4 weeks) SAQ training programme on cognitive and physical performance in preadolescent soccer players, with respect to a soccer-specific training programme based on SSG. The main finding was that 4 weeks of SAQ training programme showed comparable improvements in inhibitory control and perceptual speed with respect to SSG training programme. Moreover, confirming our hypothesis, the similar improvements (main effect of time) in cognitive performance of the SAQ and SSG group are also reflected in improvements in physical performance (5 m sprint). These findings demonstrated that activities based on non-sport-specific drills aimed at stimulating speed, agility and quickness may be as advantageous as soccer-specific activities such as SSG for stimulating both cognitive and physical domain. This suggests that SAQ training may be an alternative strategy for improving neuromuscular performance (i.e., 5 m sprint) and non-sport-specific cognitive skills (i.e., inhibitory control and perceptual speed) in preadolescents soccer players.

The importance of non-sport-specific cognitive skills for soccer performance has been widely demonstrated [6, 38–40]. For example, in a study assessing general cognitive functions, young soccer players competing at high level showed superior inhibitory control and a larger alerting effect with respect to their low-level counterparts [6]. Moreover, performance in cognitive tasks assessing executive functions was found to be associated with future success in sport in both adults [39] and young soccer players [40]. However, to the best of the authors' knowledge, apart from these cross-sectional studies, literature is lacking in studies assessing the training-associated cognitive performance changes in youths practicing soccer. One study by Alesi et al. [41] investigated the effect of a soccer training programme based on soccer-specific drills (i.e., individual skills, technique and SSG) on motor and cognitive tasks in children aged 7 to 11 years. Compared with their sedentary peers, children attending the soccer-specific

programme improved 20 m sprint, agility and selective attention [41]. The fact that our findings seem not in agreement with those found by Alesi et al. [41] may be attributed to important differences in the experimental design. The two training programmes performed in the present study were non-soccer-specific training programme (SAQ training) on one side and soccer-specific training programme (SSG training) on the other side, the latter similar to that proposed by Alesi et al. [41]. Indeed, it should be highlighted that the present lack of interaction with the concurrent main effects of time revealed general improvements in both 5 m sprint, inhibitory control and perceptual speed, demonstrating the efficacy of both SAQ and SSG for improving physical and cognitive domain. However, when looking at the *post-hoc* comparisons, the SAQ group was the only showing significant difference in both physical and cognitive performances (Figs 3–5). Specifically, the SAQ group improved its performance in the 5 m sprint (*large* ES), reaction time of the congruent and incongruent condition of the Flanker task (*large* ES), and reaction time of 15 and 20 items of the visual search task (*large* ES). The lack of significant pre-post improvements in the 5 and 10 items of the visual search task may be explained considering previous literature on the exercise-cognition interaction. Accordingly, tasks requiring greater amounts of interference control (such as incongruent condition of the Flanker task and 15 and 20 items of the visual search task) were found to be superior for revealing possible cognitive improvements related to physical activity and exercise [42, 43]. In agreement with the cognitive stimulation hypothesis, according to which the positive effects of exercise on cognition may be explained by the cognitive engagement of exercise [21, 44], the efficacy of both groups in terms of cognitive improvement could be ascribed to this notion. It is plausible that the agility activities based on repetitive cognitive *stimuli* might have contributed to maintain a cognitive engagement during the SAQ tasks, as for the SSG tasks, thus favoring benefits on non-sport-specific cognitive skills. These notions were supported by neurophysiological studies showing underlying neural mechanisms that contributed to explain the relationship between exercises with some forms of cognitive engagement and cognition [25–27, 45]. As a matter of fact, it has been demonstrated that a training programme based on coordinative exercises (engaging cognition) induced a decreased activation of the prefrontal areas when performing an executive control task (fewer resources needed to perform the task assessing inhibitory control) with a concurrent increased activation of task-specific areas involving executive and visual-spatial processing [27]. Moreover, prefrontal *cortex* (important for executive functions) and *cerebellum* (important for complex movements) were found to be activated during both motor and cognitive tasks [45, 46]. These notions lead to the conclusion of an existing association between complex motor activities (as those of SAQ and SSG) and cognitive performance [46].

A side finding of this study is the general improvement of both groups in the 5 m sprint (main effect of time). The ability to maximally run over short distances is a crucial component of the performance during soccer games [47]. Our finding agrees with those reported in previous studies showing the benefits of a SAQ training programme for improving acceleration over 5 m (5 m sprint), rather than maximal speed (20 m sprint [12, 16, 17]). The significant pre-post improvement of the SAQ group in the 5 m sprint ($P = 0.013$; *large* ES) observed in the present study may be related to the specificity of quickly foot exercises requiring short contact time, eliciting higher forces production at faster rates and resulting in an improvement in acceleration performance [48]. The lack of improvement in the COD performance (COD90) for both SAQ and SSG may be attributed to the nature of the tests itself (e.g., relatively long duration), which is mainly based on athletes' physical capacity and age. According to Lloyd et al. [49], CODs trainability is *maximum* at 13–14 years of age. Thus, potential neuromuscular adaptations derived from SAQ training may not have been adequate to affect CODS performance in the present under-10 soccer players. This seems to support the fact that focusing on CODs development in so young players may not be a primary aim of youth training

programmes as previously suggested [49]. Although the number of repetitions of sprint and changes of direction were not quantified, the literature also supports the notion that the addition of SAQ-related drills (i.e., endurance and speed training sessions) to SSG has similar effects to well-organized SSG alone for improving sprint performance [50].

## Limitations of this study

The current study presents some limitations that should be acknowledged. First, as cognitive functions develop from early childhood into adulthood [51], it is not possible to completely exclude the possibility that the improvements observed in cognitive performance (main effect of time) might be related—at least partially—to the natural development of executive functions. In this regard, there should be a control group to assess the investigated training strategies compared with no-intervention. Moreover, the training intervention lasted 4 weeks, which is a relatively short period within the training-related literature. Although we were able to observe physical and cognitive improvements after 4 weeks only, further studies should consider longer training interventions and investigate whether the improvements related to the training programmes could be also maintained after a retention period. Furthermore, measures test-retest reliability should be assessed to isolate the repeated-measure natural biological error from the differences due to the training protocol. Second, although this study employed a sample size in line with the team-sports-related literature, the relative low sample size limits the interpretation of the results as well as their generalization. Further studies employing larger sample size are necessary. Third, cognitive performance was assessed administering two computer tasks. Further but more ecological research could instead make use of some soccer-specific cognitive performance assessment (e.g., passing choice and pitch area coverage), as well as assessing the cognitive engagement of training regimes. Finally, we were not able to depict a portrait of the functional neural adaptations that potentially occurred after the 4 weeks training intervention during the execution of the cognitive tasks. Further studies could use the current experimental paradigm while measuring functional neuroimaging activities during cognitive tasks to effectively establish whether changes in behavioral outcomes would be also accompanied by changes in brain function in response to different training interventions.

## Conclusions

In conclusion, our experiment demonstrated that a short-term (4 weeks) SAQ training programme induced similar improvements on both cognitive and physical performance with respect to a soccer-specific training programme based on SSG in preadolescent soccer players. Non-sport-specific activities targeting speed, agility and quickness combined with cognitive engagement (i.e., SAQ) should be useful strategies as soccer-specific activities (i.e., SSG) to be included within a soccer training program for promoting both physical and cognitive domain in preadolescent individuals. We do believe youth sport bodies' aim should include athletes' cognitive development, as well.

## Supporting information

**S1 File.**
(XLSX)

## Author Contributions

**Conceptualization:** Athos Trecroci, Luca Cavaggioni, Alessio Rossi, Andrea Moriondo, Giampiero Merati, Hadi Nobari, Luca Paolo Ardigò, Damiano Formenti.

**Data curation:** Athos Trecroci, Luca Cavaggioni, Alessio Rossi, Andrea Moriondo, Giampiero Merati, Hadi Nobari, Luca Paolo Ardigò, Damiano Formenti.

**Formal analysis:** Athos Trecroci, Luca Cavaggioni, Alessio Rossi, Andrea Moriondo, Giampiero Merati, Hadi Nobari, Luca Paolo Ardigò, Damiano Formenti.

**Investigation:** Athos Trecroci, Luca Cavaggioni, Alessio Rossi, Andrea Moriondo, Giampiero Merati, Hadi Nobari, Luca Paolo Ardigò, Damiano Formenti.

**Methodology:** Athos Trecroci, Luca Cavaggioni, Alessio Rossi, Andrea Moriondo, Giampiero Merati, Hadi Nobari, Luca Paolo Ardigò, Damiano Formenti.

**Project administration:** Athos Trecroci, Luca Cavaggioni, Alessio Rossi, Andrea Moriondo, Giampiero Merati, Hadi Nobari, Luca Paolo Ardigò, Damiano Formenti.

**Resources:** Athos Trecroci, Luca Cavaggioni, Alessio Rossi, Andrea Moriondo, Giampiero Merati, Hadi Nobari, Luca Paolo Ardigò, Damiano Formenti.

**Software:** Athos Trecroci, Luca Cavaggioni, Alessio Rossi, Andrea Moriondo, Giampiero Merati, Hadi Nobari, Luca Paolo Ardigò, Damiano Formenti.

**Supervision:** Athos Trecroci, Luca Cavaggioni, Alessio Rossi, Andrea Moriondo, Giampiero Merati, Hadi Nobari, Luca Paolo Ardigò, Damiano Formenti.

**Validation:** Athos Trecroci, Luca Cavaggioni, Alessio Rossi, Andrea Moriondo, Giampiero Merati, Hadi Nobari, Luca Paolo Ardigò, Damiano Formenti.

**Visualization:** Athos Trecroci, Luca Cavaggioni, Alessio Rossi, Andrea Moriondo, Giampiero Merati, Hadi Nobari, Luca Paolo Ardigò, Damiano Formenti.

**Writing – original draft:** Athos Trecroci, Luca Cavaggioni, Alessio Rossi, Andrea Moriondo, Giampiero Merati, Hadi Nobari, Luca Paolo Ardigò, Damiano Formenti.

**Writing – review & editing:** Athos Trecroci, Luca Cavaggioni, Alessio Rossi, Andrea Moriondo, Giampiero Merati, Hadi Nobari, Luca Paolo Ardigò, Damiano Formenti.

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
