## [Decision Letter · Decision Letter 0]

7 Jul 2022

PONE-D-22-14492Effects of speed, agility and quickness training programme on cognitive and physical performance in young soccer playersPLOS ONE

Dear Luca Paolo Ardigò,

Thank you for submitting your manuscript to PLOS ONE.  

I have completed my evaluation of your manuscript. The reviewers recommend reconsideration of your manuscript following major revision and modification. I invite you to resubmit your manuscript after addressing the comments below.

When revising your manuscript, please consider all issues mentioned in the reviewers' comments carefully: please outline every change made in response to their comments and provide suitable rebuttals for any comments not addressed. Please note that your revised submission may need to be re-reviewed.

PLOS ONE values your contribution and I look forward to receiving your revised manuscript.￼  

We look forward to receiving your revised manuscript.

Kind regards,

Leonardo de Sousa Fortes, Ph.D.

Academic Editor

PLOS ONE

Journal Requirements:

This work is supported by the European Community’s H2020 Program under the Funding Scheme H2020-INFRAIA-2019-1 Research Infrastructures Grant Agreement 871042, www.sobigdata.eu, accessed on 2 November 2021, SoBigData++: European Integrated Infrastructure for Social Mining and Big Data Analytics. The funders had no role in study design, data collection and analysis, decision to publish, or preparation of the manuscript.

 The authors received no specific funding for this work.

3.  Please upload a copy of Figure 5  to which you refer in your text on pages 14 and 16. If the figure is no longer to be included as part of the submission please remove all reference to it within the text.

Reviewers' comments:

Reviewer's Responses to Questions

**Comments to the Author**

1. Is the manuscript technically sound, and do the data support the conclusions?

Reviewer #1: Partly

Reviewer #2: Partly

2. Has the statistical analysis been performed appropriately and rigorously? 

Reviewer #1: No

Reviewer #2: No

3. Have the authors made all data underlying the findings in their manuscript fully available?

Reviewer #1: No

Reviewer #2: Yes

4. Is the manuscript presented in an intelligible fashion and written in standard English?

Reviewer #1: Yes

Reviewer #2: Yes

5. Review Comments to the Author

Reviewer #1: Comments to the authors

Dear Authors, thank you for the opportunity to review this manuscript. Overall, I commended the author for the well-written manuscript and the interesting research topic in young soccer players. However, I have some concerns about the data interpretation and made some comments that need to be addressed before further consideration. I hope that my comments can help to improve the quality of the manuscript. My comments and suggestions can be found below.

Introduction

Page 2. Line 49-51. I understand the author's point of view, but it would be relevant to made this assumption with more caution since to date there is no direct link between computer tasks relying on executive function on specific-sport performance.

Page 4. Line 92-96. This sentence is quite confusing. Please be more concise in this statement. For example, “We decided to examine the effects of SAQ compared to sport-specific training (SSG) on [….]”.

Page 5. Line 102-104. Please provide a citation to support the hypothesis of the study. Also, there is insufficient data in the literature to date to show the superiority of the SAQ protocol compared to the SSG. Based on the data presented by the authors, I am not sure about that. Furthermore, the results reported in the current study do not seem to clearly support this hypothesis (see my comments below). Please review it.

Methods

Page 6. Line 107. Please be more specific about the participant’s characteristics (i.e., range of age along with the mean and SD reported). There is any attempt made by the authors to consider the maturation status?

Page 6. Line 109. Please standardize the term throughout the manuscript to refer to young players. Sometimes appears “children”, “preadolescent” or “young players”.

Page 6. Line 112. It is important to provide more details about the randomization process. Who did the randomization (blinded or not)? How were they assigned (1x1, counterbalanced)? Where was the random number estimated? Were all players recruited at the same time?

Page 6. Line 112. Please separate the symbols from the number. Check them throughout the manuscript.

Page 6. Line 126. Was the post-test performed immediately after the end of the training intervention, or was there a rest period in between?

Page 7. Dependent variables. Why did the authors not examine the test-retest reliability of the measures? It is important to better understand whether the differences are related to the training protocol or natural biological error in repeated measures. This must be considering a limitation of the study.

Page 8. Line 165. On what type of surface were the protocols performed?

Page 8. Line 167. Please consider change S90 to COD90. IMHO, it is unnecessary to add this unusual abbreviation.

Page 8. Line 167. How much time is between the warm-up protocol and the sprint test?

Did participants begin the test immediately behind the pair of photocells or at a distance of 0.50-1.0 m to break inertia?

Page 9. Line 177. Why was this protocol chosen? Because of the speed component in the task, it is difficult to isolate the COD ability? Also, since the authors wanted to examine cognitive performance, it is hard for me to understand why the agility component was not assessed. Reading the introduction, this is the first protocol I expected to see in the study. It is a more ecological test that includes a cognitive component in addition to the motor tasks, rather than just using computer task. This needs to be addressed in the manuscript.

Page 9. Line 188. It is critical to better describe the training intervention to allow replication of the study, especially for soccer coaches working with young players. There are several variables that need to be added to this section to make it work.

1. How intensity was defined and increased, since sprints are expected to be performed at maximum effort. If the overload was applied based on the complexity of the exercise, this needs to made clearer.

2. The description of SSG drills has to be considerably improved. There are several constraints that can be manipulated during SSG tasks to increase physical and cognitive loads (pitch size, area relative per player, rules, etc…). Based on the design of the SSG reported, it is not surprise the lack of improvement in sprints performance. Please provide more details why this SSG configuration were selected to justify why it would be expected an increase in both physical and cognitive performance. As it stands, it is very difficult to have any information about the task.

3. Verbal encouragement and feedback were used in both training interventions to encourage and motivate the participants throughout the activities. This is an important consideration to take into account, particularly to reduce between-subjects variability due to lack of engagement during SSG and sprints drills.

Page 12. Line 232. It is important to provide further information about the effect size calculation. If the intra-groups pre-to-post effects were calculated, the authors must refer to the equation provided in Dankel and Loenneke (2021) to take into account within-subjects variability.

Please see: Dankel, S. J., & Loenneke, J. P. (2021). Effect sizes for paired data should use the change score variability rather than the pre-test variability. The Journal of Strength & Conditioning Research, 35(6), 1773-1778.

Results

Figure 5 is missing from my reviewer's version.

This section is one of the primary concerns I have with this manuscript. As a result, the data interpretation will have an impact on the study's discussion and conclusion. Therefore, it is critical to carefully evaluate it in order to be more conservative about the findings.

Cognitive performance:

As mentioned by the authors, there were no interaction effects in any of the cognitive tasks evaluated. However, in the discussion section, they mentioned that “SSG. The main finding was that 4 weeks of SAQ training programme showed higher improvements in inhibitory control and perceptual speed with respect to SSG training programme” (Page 15, line 311-313).

This statement is invalid due to the lack of interaction effects. Furthermore, while there was a significant time effect for SAQ but not for SSG group in some cases, when we analyzed the pre-to-post effect size, the results were relatively similar in practically all variables evaluated. It is important to read my earlier comment about calculating the effect size. Overall, this raises serious doubts about the SAQ's superiority over the SSG, as claimed by the authors. A more conservative interpretation, in my opinion, should be explored, and the writers should be warned about it in the discussion section.

Physical performance:

There were no significant time x group interactions for any of the sprint performances studied, but the authors stated that SAQ induced greater adaptations than SSG in the 5-m sprint performance. Please note that the pre-to-post effect size is comparable across groups (SAQ = 0.64, medium; SSG = 0.70, medium). Furthermore, the test's p-value approached the significance level (p = 0.09). So, in my opinion, the lack of statistical significance may be attributed to type II error due to the smaller sample size and lack of power. This reinforces my statement that a more conservative interpretation of the data is necessary.

Discussion

Page 15. Line 311-315.

As mentioned in my previous comment, this sentence needs to be carefully reviewed. In addition, the lack of a control groups must be addressed as a limitation of the study, especially in studies including young soccer players due to the natural biological development. Therefore, in a group level it is difficult to confirm that one group is superior to other due to the lack of interaction neither that both training strategies are effective because of the lack of a control group. These questions needs to considered by the authors throughout the discussion section.

The authors can explore the differences expected after the SSG designed. It would be expected meaningful differences in sprint performance after the SSG configuration proposed. There were sufficient stimuli to increase physical and cognitive loads based on the SSG constraints?

Given my previous comments on the data interpretation, I recommend the authors to review it. As a result, considerable revisions in this section as well as the conclusion section must be addressed. Therefore, I did not provide any additional remarks in this section because I believe that significant adjustments are required.

Limitations of the study

The lack of test-retest and a control group must be considered as limitations of the study.

Conclusion

The conclusion needs to be more specific. Again, the authors stated that SAQ is superior in both cognitive and physical performance than SSG (in all parameters – this is not aligned with the results).

References

The capital letters of the titles need to be reviewed.

Reviewer #2: 

The aim of the present study was to investigate the effect of a short-term (4 weeks) non-soccer-specific training program based on speed, agility and quickness (SAQ) and a soccer-specific training program based on small-sided games (SSG) on cognitive and physical performance in soccer players. The authors conclude that SAQ improves both cognition and physical abilities more than SSG.

General comments

The article is well written and easy to follow. While intro and methods are easy to understand and I have minor suggestion, I have some major comments in the results and discussion sections, which are presented below.

The authors contextualize and highlight the importance of agility in team sports, and also the importance of cognition in team sports as it is incorporated in the concept of agility. However, I wondered if agility is important, why have the authors analyzed COD when there is no need to respond to a stimulus? Does COD is influenced by the improvement in cognition?

In addition, the presentation of the results and the interpretation in the discussion section is misleading. The authors first report there is no interaction, but then base part of the discussion on the superiority of the SAQ group over SSG, which is not supported by the statistical results.

Abstract

The abstract is well written and easy to follow. However, the results and conclusions are not supported by the authors’ findings, as in the manuscript, they report that there were no interactions in none of the variables. In L29-33 – the authors state that SAQ improved in some variables compared to SSG, which is not true based on their statistical results.

Introduction.

I have minor suggestions in this section.

1) Shorten the introduction

2) Reduce the explanation of physiological mechanisms. I was expecting that the manuscript would investigate mechanisms. It is possible that if I had this expectation, others may have it too.

Methods

L172- Why did you choose to place the timing gates at 0.6m, which is knee height, not hip height as usually it is placed?

L174 - 5-m and 20-m test . Also, describe the test better.

L177 – 90°

L228-229 – The lack of difference does not indicate that athletes were split into two groups, but rather that there were no differences between groups.

Results

If there was not interaction, but there were main effects, authors should report it. I am not convinced that the simple effect is relevant. I understand when authors report simple effects, but the interpretation requires caution, as these effects do not mean that groups were different or the one group improved more than the other. It is my understanding from reading the manuscript that authors discuss their results based on the superiority of SAQ over SSG, which is not supported by their findings.

Although the authors reported no difference at the beginning of the study. Have the authors considered using ANCOVA having the pre values as covariates instead of ANOVA? There is body of evidence suggesting that ANCOVA may be an alternative (https://pubmed.ncbi.nlm.nih.gov/16895814/).

Discussion

L332 – sprint

As I mentioned in the results section, the interpretation of the findings is problematic, as it was made on the bases of superiority of SAQ, which is not supported by statistical results.

Also, I suggest incorporating some discussion on why SAQ improved and SSG did not, based on the characteristics of training. I understand that athletes respond to a stimulus during SAQ training, but aren’t players supposed to respond to different stimuli during SSG? For example, in L362-363, authors stat that complex motor activities are related to cognitive performance. What is more complex, semi-pre-determined exercises such as those used in SAQ, or a soccer game? From this information, I would expect that players in SSG would improve more than in SSG. Can authors provide evidence that engagement is higher during SAQ, or similar tasks, than in SSG?

During the discussion on physical tasks, I suggest that authors include some characteristics of the training. For instance, do players perform sprint and change of directions during SSG? How many repetitions on average? Do these change-of-direction tasks and sprint are performed at maximal effort during SSG? Maybe SSG did not improve as much as SAQ due to lower number of repetitions. https://pubmed.ncbi.nlm.nih.gov/34079163/

https://pubmed.ncbi.nlm.nih.gov/34079175/

6. PLOS authors have the option to publish the peer review history of their article (what does this mean?). If published, this will include your full peer review and any attached files.

Reviewer #1: No

Reviewer #2: No

---

## [Author Response · Author response to Decision Letter 0]

23 Sep 2022

Response to Editor Comments

Point 1: 1. Please ensure that your manuscript meets PLOS ONE's style requirements, including those for file naming. The PLOS ONE style templates can be found at 

Response 1: We thank expert editor for his suggestion. Figures were saved as *.tif files.

Point 2: We note that you have provided funding information that is not currently declared in your Funding Statement. However, funding information should not appear in the Acknowledgments section or other areas of your manuscript.

Please remove any funding-related text from the manuscript and let us know how you would like to update your Funding Statement.

Please include your amended statements within your cover letter.

Response 2: Acknowledgments section was removed from manuscript. Formerly provided funding information was added to cover letter.

Point 3: Please upload a copy of Figure 5 to which you refer in your text on pages 14 and 16.

Response 3: We apologize for the error. Figure 5 was added.

Point 4: We note that you have indicated that data from this study are available upon request. PLOS only allows data to be available upon request if there are legal or ethical restrictions on sharing data publicly.

Response 4: We thank expert editor for his suggestion. Raw data were added as S1_file.xlsx.

We hope that the manuscript has now reached the standard necessary for formal acceptance endorsement in PLOS ONE.

Best regards

 

Response to Reviewer 1 Comments

Point 1: Page 2. Line 49-51. I understand the author's point of view, but it would be relevant to made this assumption with more caution since to date there is no direct link between computer tasks relying on executive function on specific-sport performance.

Response 1: We thank the expert reviewer for her/his suggestion. Assumption was “smoothed” as follows:

“These findings suggest that non-sport-specific cognitive skills such as cognitive control may be related to performance, especially in team sports.”

Point 2: Page 4. Line 92-96. This sentence is quite confusing. Please be more concise in this statement. For example, “We decided to examine the effects of SAQ compared to sport-specific training (SSG) on [….]”.

Response 2: Following your suggestion, sentence was more coincise as follows:

“We decided to examine the effects of SAQ compared with sport-specific training (i.e., small-sided games, SSG) on physical and cognitive performances in young soccer players.”

Point 3: Page 5. Line 102-104. Please provide a citation to support the hypothesis of the study.

Response 3: Citations were provided to support the hypothesis of the study as follows:

“Based on results detected in different populations regarding the association between physical and cognitive abilities [4-7] and – at least as preliminary findings – the mutual beneficial effects of physical and cognitive training [7,12,24,26,29], we hypothesized that training based on SAQ would be more effective than a mere sport-specific training for improving cognitive and physical performance in preadolescents soccer players.”

Point 4: Also, there is insufficient data in the literature to date to show the superiority of the SAQ protocol compared to the SSG. Based on the data presented by the authors, I am not sure about that. Furthermore, the results reported in the current study do not seem to clearly support this hypothesis (see my comments below). Please review it.

Response 4: Please, read specific answers below.

Point 5: Page 6. Line 107. Please be more specific about the participant’s characteristics (i.e., range of age along with the mean and SD reported). There is any attempt made by the authors to consider the maturation status?

Response 5: Age range was indicated. We believe that the lack of maturation status assessment may not represent a limitation because our participants were pre-pubertal (age range 9-11 years). In fact, growth spurt normally occurs in 12-14 years in males (Reilly, 2004).

Point 6: Page 6. Line 109. Please standardize the term throughout the manuscript to refer to young players. Sometimes appears “children”, “preadolescent” or “young players”.

Response 6: “preadolescent” term was adopted throughout MS.

Point 7: Page 6. Line 112. It is important to provide more details about the randomization process. Who did the randomization (blinded or not)? How were they assigned (1x1, counterbalanced)? Where was the random number estimated? Were all players recruited at the same time?

Response 7: The randomization process was based on a simple randomization procedure. We did not need in our case to operate a stratified randomization because the initial group of recruited subjects was highly homogeneous for demographic and anthropometric characteristics and for training level (indeed, the 2 randomized groups resulted to be well matched for any of the main anthropometric and demographic values). The randomization procedure was performed by a blinded external operator, by using a software-generated random series of numbers. All players were recruited at the same time.

Point 8: Page 6. Line 112. Please separate the symbols from the number. Check them throughout the manuscript.

Response 8: Symbols were separated from numbers throughout MS.

Point 9: Page 6. Line 126. Was the post-test performed immediately after the end of the training intervention, or was there a rest period in between?

Response 9: Yes. Sentence was accordingly changed as follows:

“After the four weeks of training (namely during the first week thereafter), participants underwent post-test sessions that were identical to the pre-test.”

Point 10: Page 7. Dependent variables. Why did the authors not examine the test-retest reliability of the measures? It is important to better understand whether the differences are related to the training protocol or natural biological error in repeated measures. This must be considering a limitation of the study.

Response 10: Lack of measures test-retest reliability was acknowledged as study limitation as follows:

“Furthermore, measures test-retest reliability should be assessed to isolate the repeated-measure natural biological error from the differences due to the training protocol.”

Point 11: Page 8. Line 165. On what type of surface were the protocols performed?

Response 11: Information was provided.

Point 12: Page 8. Line 167. Please consider change S90 to COD90. IMHO, it is unnecessary to add this unusual abbreviation.

Response 12: “S90” was changed to “COD90” throughout MS.

Point 13: Page 8. Line 167. How much time is between the warm-up protocol and the sprint test?

Did participants begin the test immediately behind the pair of photocells or at a distance of 0.50-1.0 m to break inertia?

Response 13: Information was provided.

Point 14: Page 9. Line 177. Why was this protocol chosen? Because of the speed component in the task, it is difficult to isolate the COD ability? Also, since the authors wanted to examine cognitive performance, it is hard for me to understand why the agility component was not assessed. Reading the introduction, this is the first protocol I expected to see in the study. It is a more ecological test that includes a cognitive component in addition to the motor tasks, rather than just using computer task. This needs to be addressed in the manuscript.

Response 14: We tried to address your issues throughout MS as follows:

“Validity and reliability of COD90 were previously reported [11]. Namely, Sporis et al. found out COD90 reliability to be the highest over six different agility tests [11] (in Sprint with ninety degrees turns). Third, cognitive performance was assessed administering two computer tasks. Further but more ecological research could instead make use of some soccer-specific cognitive performance assessment (e.g., passing choice and pitch area coverage). (in Limitations of this study)”.

We would like to highlight that to date, it is well-known that no gold standard agility assessments exist, especially for age as our sample. Moreover, we previously observed the efficacy of a SAQ protocol for improving both physical (sprint) and agility (reactive agility test) performance in young soccer players (Trecroci et al., 2016). In the present case, we were interested in studying whether there would be an effect also on the mere cognitive component of agility (using computer-based tasks assessing basic cognitive functions), regardless the ecological context.

Point 15: Page 9. Line 188. It is critical to better describe the training intervention to allow replication of the study, especially for soccer coaches working with young players. There are several variables that need to be added to this section to make it work.

1. How intensity was defined and increased, since sprints are expected to be performed at maximum effort. If the overload was applied based on the complexity of the exercise, this needs to made clearer.

Response 15: Intensity was defined and increased based on Borg's rate of perceived exertion scale (running from 0 to 10). Overload application over both SAQ and SSG training programmes is shown in Table 1.

Point 16: 2. The description of SSG drills has to be considerably improved. There are several constraints that can be manipulated during SSG tasks to increase physical and cognitive loads (pitch size, area relative per player, rules, etc…). Based on the design of the SSG reported, it is not surprising the lack of improvement in sprints performance. Please provide more details why this SSG configuration were selected to justify why it would be expected an increase in both physical and cognitive performance. As it stands, it is very difficult to have any information about the task.

Response 16: Thank you for this suggestion. We provided a detailed description of the SSG tasks to increase physical and cognitive loads within Table 1. We believe that now the SSG description is more complete providing the Readers a better comprehension of the training tasks.

Point 17: 3. Verbal encouragement and feedback were used in both training interventions to encourage and motivate the participants throughout the activities. This is an important consideration to take into account, particularly to reduce between-subjects variability due to lack of engagement during SSG and sprints drills.

Response 17: Information was provided.

Point 18: Page 12. Line 232. It is important to provide further information about the effect size calculation. If the intra-groups pre-to-post effects were calculated, the authors must refer to the equation provided in Dankel and Loenneke (2021) to take into account within-subjects variability.

Please see: Dankel, S. J., & Loenneke, J. P. (2021). Effect sizes for paired data should use the change score variability rather than the pre-test variability. The Journal of Strength & Conditioning Research, 35(6), 1773-1778.

Response 18: We thank the Reviewer for the suggestion. We have now calculated the intra-group pre-to-post effect sizes by using the equation by Dankel and Loenneke (2021).

Point 19: Figure 5 is missing from my reviewer's version.

Response 19: We apologize for the error. Figure 5 was added.

Point 20: As mentioned by the authors, there were no interaction effects in any of the cognitive tasks evaluated. However, in the discussion section, they mentioned that “The main finding was that 4 weeks of SAQ training programme showed higher improvements in inhibitory control and perceptual speed with respect to SSG training programme” (Page 15, line 311-313).

This statement is invalid due to the lack of interaction effects. Furthermore, while there was a significant time effect for SAQ but not for SSG group in some cases, when we analyzed the pre-to-post effect size, the results were relatively similar in practically all variables evaluated. It is important to read my earlier comment about calculating the effect size. Overall, this raises serious doubts about the SAQ's superiority over the SSG, as claimed by the authors. A more conservative interpretation, in my opinion, should be explored, and the writers should be warned about it in the discussion section.

Response 20: We thank the Reviewer for this comment. It allowed us to interpret with caution the statistical analysis. As suggested, we have re-calculated and added effect sizes in the Results section, and particular care has been devoted to discuss the lack of interaction and the main effect of time, rather than focusing only on post-hoc tests. Indeed, Discussion has been modified in accordance with a more appropriate interpretation of the statistical analysis, devoting importance to the absence of interactions, and to the significance of the main effects of time. Accordingly, conclusions derived by the data have been cautiously exposed, both in the abstract and in the text, using a more conservative interpretation.

Point 21: There were no significant time x group interactions for any of the sprint performances studied, but the authors stated that SAQ induced greater adaptations than SSG in the 5-m sprint performance. Please note that the pre-to-post effect size is comparable across groups (SAQ = 0.64, medium; SSG = 0.70, medium). Furthermore, the test's p-value approached the significance level (p = 0.09). So, in my opinion, the lack of statistical significance may be attributed to type II error due to the smaller sample size and lack of power. This reinforces my statement that a more conservative interpretation of the data is necessary.

Response 21: Please refer to Response 20.

Point 22: Page 15. Line 311-315.

As mentioned in my previous comment, this sentence needs to be carefully reviewed. … Therefore, in a group level it is difficult to confirm that one group is superior to other due to the lack of interaction neither that both training strategies are effective because of the lack of a control group. These questions need to be considered by the authors throughout the discussion section.

The authors can explore the differences expected after the SSG designed. It would be expected meaningful differences in sprint performance after the SSG configuration proposed. There were sufficient stimuli to increase physical and cognitive loads based on the SSG constraints?

Given my previous comments on the data interpretation, I recommend the authors to review it. As a result, considerable revisions in this section as well as the conclusion section must be addressed. Therefore, I did not provide any additional remarks in this section because I believe that significant adjustments are required.

Response 22: We thank the Reviewer for raising such concern. Accordingly, we have modified the whole text explaining better the SSG protocol. Specifically, Table 1 has been revised by including a detailed description of the SSG protocol, allowing the Reader to have a straightforward understanding of the physical and cognitive stimuli of SSG. Parallelly, giving importance to the lack of interaction and to the literature, we have decided to be more conservative also on our hypothesis.

Point 23: In addition, the lack of a control groups must be addressed as a limitation of the study, especially in studies including young soccer players due to the natural biological development. Therefore, in a group level it is difficult to confirm that one group is superior to other due to the lack of interaction neither that both training strategies are effective because of the lack of a control group.

Response 23: Lack of controls was acknowledged as a limitation of the study as follows:

“Or, at least, there should be a control group to assess the investigated training strategies compared with no-intervention.”

Point 24: The lack of test-retest and a control group must be considered as limitations of the study.

Response 24: The lack of control group and test-retest reliability assessment were acknowledged as limitations of the study (read above).

Point 25: The conclusion needs to be more specific. Again, the authors stated that SAQ is superior in both cognitive and physical performance than SSG (in all parameters – this is not aligned with the results).

Response 25: We thank the Reviewer. Conclusions were modified by adding a more cautious and rigorous interpretation of the results derived by the inferential statistics.

Point 26: The capital letters of the titles need to be reviewed.

Response 26: We thank expert reviewer for his suggestion. The capital letters of the titles were reviewed. Actually some of them were mostly capitalized (source PubMed).

We hope that the manuscript has now reached the standard necessary for formal acceptance endorsement in PLOS ONE.

Best regards

 

Response to Reviewer 2 Comments

Point 1: … I wondered if agility is important, why have the authors analyzed COD when there is no need to respond to a stimulus? Does COD is influenced by the improvement in cognition?

Response 1: We thank the expert reviewer for her/his suggestion. In the literature, change of direction ability with 90° turns is considered a (valid and reliable) agility test. This was stated as follows:

“Validity and reliability of COD90 were previously reported [11]. Namely, Sporis et al. found out COD90 reliability to be the highest over six different agility tests [11].”

We would like to highlight that to date, it is well-known that no gold standard agility assessments exist, especially for age as our sample. Moreover, we previously observed the efficacy of a SAQ protocol for improving both physical (sprint) and agility (reactive agility test) performance in young soccer players (Trecroci et al., 2016). In the present case, we were interested in studying whether there would be an effect also on the mere cognitive component of agility (using computer-based tasks assessing basic cognitive functions), regardless the ecological context (although we recognized its importance).

Point 2: … The authors first report there is no interaction, but then base part of the discussion on the superiority of the SAQ group over SSG, which is not supported by the statistical results.

Response 2: We thank the Reviewer for this comment. It allowed us to interpret with caution the statistical analysis. As suggested, we have re-calculated added effect sizes in the Results section, and particular care has been devoted to discuss the lack of interaction and the main effect of time, rather than focusing only on post-hoc tests. Indeed, Discussion has been modified in accordance with a more appropriate interpretation of the statistical analysis, devoting importance to the absence of interactions, and to the significance of the main effects of time. Accordingly, conclusions derived by the data have been cautiously exposed, both in the abstract and in the text, using a more conservative interpretation.

Point 3: Abstract

The abstract is well written and easy to follow. However, the results and conclusions are not supported by the authors’ findings, as in the manuscript, they report that there were no interactions in none of the variables. In L29-33 – the authors state that SAQ improved in some variables compared to SSG, which is not true based on their statistical results.

Response 3: The Reviewer is right. The abstract has now been modified in accordance with an appropriate interpretation of the results.

Point 4: Shorten the introduction.

Reduce the explanation of physiological mechanisms. I was expecting that the manuscript would investigate mechanisms. It is possible that if I had this expectation, others may have it too.

Response 4: We thank the Reviewer for this suggestion. The Introduction has been now shortened deleting the description of the effects of exercise on brain structures in animals.

Point 5: L172- Why did you choose to place the timing gates at 0.6m, which is knee height, not hip height as usually it is placed?

Response 5: Timing gates were placed at 0.6 m of height above the ground to cope with participants’ height and posture during sprint initial phase. Sentence was extended as follows:

“In line with participants’ (limited) ~10-yr age height and allowing a usual a little bit crouched posture during initial acceleration, the timing gates were placed at only 0.60 m above the ground.”

Point 6: L174 - 5-m and 20-m test. Also, describe the test better.

Response 6: Thank you. Although the simplicity of the tests does not require numerous details, the description of the tests have now been extended: “They were requested to accelerate from the starting line and to run as fast as possible until the end line. After the end of a trial, participants were asked to return to the starting line by walking slowly.”

Point 7: L177 – 90°.

Response 7: Suggestion was operated.

Point 8: L228-229 – The lack of difference does not indicate that athletes were split into two groups, but rather that there were no differences between groups.

Response 8: Sentence was changed as follows:

“No significant difference between groups – as detected by means of unpaired Student’s t-test – was found for each variable in pretraining test evaluation confirming the appropriateness of initial random splitting all participants into the two intervention groups.”

Point 9: If there was not interaction, but there were main effects, authors should report it. I am not convinced that the simple effect is relevant. I understand when authors report simple effects, but the interpretation requires caution, as these effects do not mean that groups were different or the one group improved more than the other. It is my understanding from reading the manuscript that authors discuss their results based on the superiority of SAQ over SSG, which is not supported by their findings.

Although the authors reported no difference at the beginning of the study. Have the authors considered using ANCOVA having the pre values as covariates instead of ANOVA? There is body of evidence suggesting that ANCOVA may be an alternative (https://pubmed.ncbi.nlm.nih.gov/16895814/).

Response 9: We thank the Reviewer for this comment. As already stated, we have interpreted results with more caution throughout the whole manuscript. Regarding ANCOVA, although this could be an alternative approach, based on the lack of differences in the baseline, we have decided to maintain our approach, keeping the analysis simple. We believe that a simpler approach could be more understandable also by the Readers being more straightforward.

Point 10: L332 – sprint.

Response 10: Mistake was amended.

Point 11: As I mentioned in the results section, the interpretation of the findings is problematic, as it was made on the bases of superiority of SAQ, which is not supported by statistical results.

Response 11: Please refer to Response 2.

Point 12: Also, I suggest incorporating some discussion on why SAQ improved and SSG did not, based on the characteristics of training. I understand that athletes respond to a stimulus during SAQ training, but aren’t players supposed to respond to different stimuli during SSG? For example, in L362-363, authors state that complex motor activities are related to cognitive performance. What is more complex, semi-pre-determined exercises such as those used in SAQ, or a soccer game? From this information, I would expect that players in SAQ would improve more than in SSG.

Response 12: In keeping with the previous comments provided by the Reviewer, along with a more conservative interpretation of the statistical results, we would not remark a plausible superiority of SAQ (although post-hoc and effect sizes are consistent) over SSG (given the lack of interaction). However, for convenience we report here evidence about the beneficial effects of SAQ on both physical (Milanovic et al., 2013; Polman et al., 2009) and agility (Trecroci et al., 2016) performance as compared with sport-specific training (as SSG). We would stress that, to the best of our knowledge, there is no evidence comparing the actual cognitive component involved in the two training types (SAQ and SSG). 

Point 13: Can authors provide evidence that engagement is higher during SAQ, or similar tasks, than in SSG?

Response 13: We would stress that, to the best of our knowledge, there is no evidence comparing the actual cognitive component involved in the two training types (SAQ and SSG). Unfortunately, we were not able to quantify the cognitive engagement of our participants during the two trainings. This was also added within the limitations’ paragraph. 

Point 14: During the discussion on physical tasks, I suggest that authors include some characteristics of the training. For instance, do players perform sprint and change of directions during SSG? How many repetitions on average? Do these change-of-direction tasks and sprint are performed at maximal effort during SSG?

Response 14: More details on the physical training have been added in the Table 1. As SSG are smaller version of a formal game, they of course include sprints and change of directions. Players were required to perform as maximum as possible during SSG. However, due to the absence of GPS-based metrics, we were not able to established the number of sprint and change of directions and whether they were performed at maximally intensity. We could only suppose that their number increased progressively by the pitch size and number of players (please refer to Table 1).

Point 15: Maybe SSG did not improve as much as SAQ due to lower number of repetitions.

https://pubmed.ncbi.nlm.nih.gov/34079163/

https://pubmed.ncbi.nlm.nih.gov/34079175/

Response 15: We thank the Reviewer for this comment. Effect sizes and post-hoc p-values of our study seem to suggest that SAQ may be superior than SSG. However, as already stated, the lack on interaction does not permit this statement. Although the number of repetitions were not quantified, the literature also supports the notion that the addition of SAQ-related drills (i.e., endurance and speed training sessions) to SSG was no more effective than well-organized SSG alone for improving sprint performance (Castillo et al., 2021). Please refer to the Discussion section.

We hope that the manuscript has now reached the standard necessary for formal acceptance endorsement in PLOS ONE.

Best regards

---

## [Decision Letter · Decision Letter 1]

2 Nov 2022

Effects of speed, agility and quickness training programme on cognitive and physical performance in preadolescent soccer players

PONE-D-22-14492R1

Dear Dr. Luca Paolo Ardigò,

We’re pleased to inform you that your manuscript has been judged scientifically suitable for publication and will be formally accepted for publication once it meets all outstanding technical requirements.

Kind regards,

Leonardo de Sousa Fortes, Ph.D.

Academic Editor

PLOS ONE

Reviewers' comments:

Reviewer's Responses to Questions

**Comments to the Author**

1. If the authors have adequately addressed your comments raised in a previous round of review and you feel that this manuscript is now acceptable for publication, you may indicate that here to bypass the “Comments to the Author” section, enter your conflict of interest statement in the “Confidential to Editor” section, and submit your "Accept" recommendation.

Reviewer #2: All comments have been addressed

2. Is the manuscript technically sound, and do the data support the conclusions?

Reviewer #2: Yes

3. Has the statistical analysis been performed appropriately and rigorously? 

Reviewer #2: Yes

4. Have the authors made all data underlying the findings in their manuscript fully available?

Reviewer #2: Yes

5. Is the manuscript presented in an intelligible fashion and written in standard English?

Reviewer #2: Yes

6. Review Comments to the Author

Reviewer #2: I commend the authors for their hard work. Also, I appreciate the clarity of their responses to my previous comments.

I have no further suggestions.

7. PLOS authors have the option to publish the peer review history of their article (what does this mean?). If published, this will include your full peer review and any attached files.

Reviewer #2: No

---

## [Editor Report · Acceptance letter]

22 Nov 2022

PONE-D-22-14492R1 

Effects of speed, agility and quickness training programme on cognitive and physical performance in preadolescent soccer players 

Dear Dr. Ardigò:

I'm pleased to inform you that your manuscript has been deemed suitable for publication in PLOS ONE. Congratulations! Your manuscript is now with our production department. 

Kind regards, 

on behalf of

Dr. Leonardo de Sousa Fortes 

Academic Editor

PLOS ONE